# Rapid Non-Destructive Detection Technology in the Field of Meat Tenderness: A Review

**DOI:** 10.3390/foods13101512

**Published:** 2024-05-13

**Authors:** Yanlei Li, Huaiqun Wang, Zihao Yang, Xiangwu Wang, Wenxiu Wang, Teng Hui

**Affiliations:** 1Mechanical and Electrical Engineering College, Beijing Polytechnic College, Beijing 100042, China; whq@bgy.edu.cn (H.W.); yzh05201900729@163.com (Z.Y.); wxw020619@163.com (X.W.); 2Modern Agricultural Engineering Key Laboratory at Universities of Education Department of Xinjiang Uygur Autonomous Region, Alaer 843300, China; 3Food Science and Technology College, Hebei Agricultural University, Baoding 071001, China; cauwwx@hebau.edu.cn; 4Food Science College, Sichuan Agricultural University, Ya’an 625014, China; huiteng@sicau.edu.cn

**Keywords:** meat, tenderness, non-destructive testing

## Abstract

Traditionally, tenderness has been assessed through shear force testing, which is inherently destructive, the accuracy is easily affected, and it results in considerable sample wastage. Although this technology has some drawbacks, it is still the most effective detection method currently available. In light of these drawbacks, non-destructive testing techniques have emerged as a preferred alternative, promising greater accuracy, efficiency, and convenience without compromising the integrity of the samples. This paper delves into applying five advanced non-destructive testing technologies in the realm of meat tenderness assessment. These include near-infrared spectroscopy, hyperspectral imaging, Raman spectroscopy, airflow optical fusion detection, and nuclear magnetic resonance detection. Each technology is scrutinized for its respective strengths and limitations, providing a comprehensive overview of their current utility and potential for future development. Moreover, the integration of these techniques with the latest advancements in artificial intelligence (AI) technology is explored. The fusion of AI with non-destructive testing offers a promising avenue for the development of more sophisticated, rapid, and intelligent systems for meat tenderness evaluation. This integration is anticipated to significantly enhance the efficiency and accuracy of the quality assessment in the meat industry, ensuring a higher standard of safety and nutritional value for consumers. The paper concludes with a set of technical recommendations to guide the future direction of non-destructive, AI-enhanced meat tenderness detection.

## 1. Introduction

Meat is an indispensable part of the human diet, offering a rich array of nutrients including proteins, lipids, trace elements, and minerals that are essential for human health and well-being [1,2]. As the living standards in the nation rise, there is an increasing demand for high-quality meat that is not only nutritious but also safe for consumption. This heightened awareness has made the quality control of meat products a critical concern for both consumers and producers.

Tenderness encompasses the resistance of meat to shearing forces and its overall texture, which is influenced by factors such as the cross-linking of connective tissues, interactions between muscle protein molecules, water content, and collagen levels [3,4,5]. The post-slaughter biochemical processes that muscles undergo can dramatically alter the tissue’s conditions, thereby impacting the meat’s quality. These changes can adversely affect consumer interests and the reputation of meat enterprises, underscoring the importance of rapid and accurate tenderness assessment in ensuring the quality and safety of meat products [6,7,8].

The current methods for assessing meat tenderness include traditional and rapid non-destructive testing approaches. Traditional testing methods, such as shear force and chemical analyses, are often time-consuming and can result in sample damage, posing challenges in meeting the demands of modern, high-throughput production environments. In contrast, improved rapid non-destructive testing techniques that leverage spectroscopy, electrical, and magnetic properties have been developed to enable swift and accurate assessments without compromising the samples’ inherent physical and chemical properties. These advancements have significantly enhanced the detection efficiency, reduced costs, and expanded the scope of their application in the rapid quality assessment of meat products [9,10,11,12].

This paper aims to provide a comprehensive overview of the recent advancements in the rapid and non-destructive detection techniques for meat tenderness. It explores the application of these techniques in various contexts, discusses the limitations and challenges associated with their use, and importantly, highlights the potential for overcoming these obstacles. By doing so, the paper seeks to offer valuable insights and inspiration for the further development and application of these cutting-edge technologies in the field of meat quality assessment.

## 2. Application Progress of the Rapid Non-Destructive Detection Technologies for Meat Tenderness

Rapid non-destructive testing technology has emerged as a cutting-edge approach in modern analytical techniques, employing advanced instruments to evaluate the external features and internal composition of samples without causing any damage. This innovative method harnesses the inherent properties of the materials being tested, such as optical, acoustic, and electromagnetic characteristics, to gain insights into the physical, mechanical, and structural attributes of the samples. The technology draws upon a diverse array of disciplines, including acoustics, optics, electrical engineering, thermodynamics, and magnetism, among others [13,14,15].

The utilization of advanced techniques such as near-infrared (NIR) spectroscopy, hyperspectral imaging (HSI), Raman spectroscopy, airflow–optical fusion detection, and nuclear magnetic resonance (NMR) technology [16,17,18] has revolutionized the way samples are analyzed. The rapid non-destructive testing technology stands out for its comprehensive capabilities, offering a wide range of applications across various industries. In the context of meat and meat products, this technology presents enormous potential for development. It enables the accurate assessment of key quality parameters, such as tenderness, freshness, and composition, without the need for labor-intensive and time-consuming traditional methods. The adoption of these technologies in the meat industry is poised to bring about significant improvements in efficiency, safety, and quality control. By providing detailed and precise information about the internal structure and composition of meat products, these non-destructive techniques can enhance the overall quality assurance processes, ensuring that consumers receive high-quality and safe products.

### 2.1. Spectroscopy Technology

Spectroscopic detection is an analytical method utilized to identify the physical properties and determine the chemical composition based on the unique spectrum of a substance. It operates on the fundamental principle of analyzing characteristic wavelengths and the intensities of light absorption, emission, or scattering to ascertain the chemical composition and material structure. This interaction between light and matter varies with wavelength, with different substances exhibiting distinct behaviors in terms of light absorption, emission, or scattering across various wavelength regions [19]. Given the intricate composition of meat products, assessing their quality presents significant challenges. However, spectral detection technology’s inherent characteristics adeptly address this challenge, rendering it a widely adopted method in meat quality assessment.

#### 2.1.1. Visible-Near Infrared (Vis-NIR) Spectroscopy

NIR spectroscopy, spanning from 700~2500 nm (14,286~4000 cm^−1^), lies between ultraviolet visible light (UV Vis) and mid-infrared light (MIR). It delineates short-wave (700~1100 nm) and long-wave (1100 cm^−1^ to 2500 nm) NIR regions [20]. Operating on the principle of molecular vibration transitions, NIR spectroscopy records frequency doubling and combined absorption, mainly of hydrogen-containing group vibrations (X-H, where X = C, N, O). This analytical method measures substance absorption, reflection, or transmission within the NIR range, facilitating precise determination of molecular structure and compound identification [21,22,23]. The common detection device and spectrum used for analysis are showed in Figure 1 and Figure 2, respectively. Due to its capability to discern hydrocarbon organic substance composition and properties, NIR spectroscopy is well-suited for various applications. 

NIR spectroscopy coupled with the successive projection algorithm (SPA) has yielded promising results in the tenderness detection of meat products. By scanning wavelengths between 900 and 1800 nm, the raw data are preprocessed using multivariate scattering correction (MSC) and modeled using partial least squares (PLS). This approach achieved a prediction accuracy of 0.96329, demonstrating the feasibility of non-destructive meat tenderness testing [24]. Furthermore, NIR spectroscopy has been leveraged for moisture content detection in pork, achieving a model prediction accuracy of 90.48% [25]. These findings underscore NIR spectroscopy’s potential in developing portable non-destructive testing equipment for tenderness evaluation.

Integrating NIR reflectance spectroscopy with visible light, Balage et al., 2015 attempted pork tenderness detection. Although achieving a prediction accuracy of 78%, challenges arose possibly due to sample composition variations impacting the light scattering effects [26]. Similarly, Wyrwisz et al. developed an online meat tenderness measurement system using NIR spectroscopy combined with fiber optic systems. Despite efforts, the model accuracy remained a concern, with occasional RPD values surpassing 2 [27]. Nonetheless, stepwise regression in the MLR method exhibited promising results, hinting at NIR spectroscopy’s adaptability to different algorithms and instruments for enhanced detection [28].

However, NIR spectroscopy’s indirect nature necessitates extensive experimental samples for model establishment, impacting calibration model applicability and data accuracy. The efficacy of calibration models and econometric method selection critically influences analysis outcomes. Moreover, NIR spectroscopy is ill-suited for analyzing dispersed samples with frequent variations. Therefore, the paramount objective in NIR spectroscopy development is establishing universal, representative calibration models to ensure data accuracy.

#### 2.1.2. Hyperspectral Imaging (HIS) Technology

HSI is an innovative non-destructive testing technology that seamlessly integrates spectroscopy, information processing, and computer vision techniques. This advanced technology is characterized by its unique “Image Spectral Integration,” which enables simultaneous acquisition of both imaging and spectral data from the test samples without needing any sample damage or preprocessing. Its benefits include high analytical efficiency, straightforward operation, and cost-effectiveness, making it a preferred choice for various applications [29,30].

The versatility of HSI technology allows for a comprehensive assessment of meat product quality. By leveraging imaging technology, it can detect the external features of meat products, while spectral technology reveals the internal quality and food safety information. This dual capability facilitates an in-depth evaluation of meat tenderness, establishing HSI as a reliable and user-friendly non-destructive testing method [31,32]. Numerous scholars worldwide have employed HSI for assessing meat tenderness, utilizing the results to evaluate the overall quality of meat products comprehensively.

In previous research, HSI technology has been effectively employed for the rapid quality assessment of meat, yielding positive outcomes [18]. Given that tenderness is a critical aspect of quality evaluation, it is feasible to use HSI technology for testing meat tenderness. Configured as depicted in Figure 3, the HSI system gathers data from samples and employs an enhanced Lorentz function to fit the light scattering curves at various wavelengths. By integrating Principal Component Analysis (PCA), a linear discriminant model is formulated, achieving a prediction accuracy of 75% in the validation set. Intriguingly, the presence of fat spots was found to have a negligible impact on the tenderness prediction accuracy when a larger diameter incident beam was used on whole steaks, highlighting a marked enhancement in system robustness [33].

Balage et al., 2018 explored the application of HSI for evaluating beef tenderness, utilizing a hyperspectral camera to scan samples across the 928~2524 nm spectrum with a spectral resolution of 6.3 nm and a spatial resolution of 10 μm. They applied Partial Least Squares combined with Discriminant Analysis (PLS-DA) for data processing and analysis, developing both local and full sample models. The findings indicated that local models, with a 72% accuracy rate, were more effective in discerning the tenderness of beef, underscoring the importance of clearly defining HSI application objectives to establish more accurate and robust models [34].

Zhao et al. established a robust connection between models and samples by extracting reflective spectral information from the samples. They employed a stepwise regression algorithm coupled with a Genetic Algorithm (GA) to identify the characteristic bands associated with the Warner–Bratzler Shear Force (WBSF) value of beef. Furthermore, they utilized PCA to extract three principal components from the samples and developed tenderness-level discrimination models based on a Support Vector Machine (SVM) and Linear Discriminant Analysis (LDA). The study revealed that the PCA-based prediction model outperformed the feature band image model, with the LDA model exhibiting higher recognition accuracy than the SVM model, achieving a discrimination accuracy of 94.44% [35]. This research offers valuable insights for analyzing the distribution of the quality characteristics in whole beef samples and comprehensively evaluating beef tenderness.

Studies leveraging HSI technology’s “Image Spectral Integration” have also been conducted on Ningxia Tan sheep meat. A hyperspectral system captured 128 images of Tan sheep meat within the 400~1000 nm range. The original spectra, when combined with the Savitzky–Golay (SG) convolutional smoothing preprocessing method, facilitated the extraction of nine characteristic wavelengths. Both Partial Least Squares Regression (PLSR) and Multiple Linear Regression (MLR) were employed for modeling and analysis. The results indicated that the PLSR model, which integrated feature wavelengths and surface fat distribution image features for predicting lamb tenderness, demonstrated superior predictive performance, with a correlation coefficient of 0.89 in the prediction set [36]. This study provides a theoretical foundation for the effective application of hyperspectral technology in lamb tenderness detection.

Recognizing the impact of refrigeration on meat product tenderness, Wang et al. utilized HSI technology to conduct non-destructive tenderness testing on chicken samples that had been refrigerated for 0~6 days. The spectral scanning range was 900~1700 nm. After acquiring the raw data, SG convolutional smoothing and baseline correction were applied for data preprocessing. A detection model was established using PLSR, and it was observed that the model, post-SG preprocessing of spectral data, exhibited the best predictive performance, with a correlation coefficient of 0.94 in the prediction set [37]. This research offers a novel approach for meat product enterprises to rapidly measure tenderness and evaluate the product quality during the production process, showcasing significant practical application potential.

Yu et al. designed a hyperspectral image acquisition system for the rapid and non-destructive detection of cold and fresh lamb meat tenderness. They employed a Minimum Spanning Tree (MSC) for spectral data preprocessing and combined PCA with a Gray Level Co-occurrence Matrix (GLCM) to extract image feature information. Both a Backpropagation Neural Network (BPNN) and SVM logarithmic data were utilized for modeling and analysis. The study found that the BPNN model provided better predictive performance than the SVM, with a correlation coefficient of 0.85 in the prediction set [38]. This research serves as a reference for researchers aiming to design portable or miniaturized hyperspectral non-destructive testing instruments for lamb tenderness.

The optimization of algorithms and enhancement of models is crucial for advancing the non-destructive testing technology in HSI. A study predicting lamb meat tenderness without damage used hyperspectral and chemometric methods. The hyperspectral spectra of lamb meat were collected across two bands, 400~1000 nm and 900~1700 nm, with the original spectral data in both bands preprocessed using MSC, derendering, baseline, SNV, normalization, and SG. SPA, Competitive Adaptive Reweighted Sampling (CARS), and Variable Combination Population Analysis (VCPA) were used, with the Interval Variable Iterative Space Shrink Approach (IVISSA) to optimize the feature wavelengths for preprocessed data. Models were developed using the PLSR algorithm, and various numerical values were compared. The study found that the OS-IVISSA-PLSSR tenderness prediction model had the highest predictive performance index, with a correlation coefficient of 0.79 for the prediction set [39]. This outcome demonstrates that model optimization can significantly reduce the computational operations, enhance the detection efficiency, and improve the model’s accuracy and stability, ultimately increasing the detection precision.

HSI technology excels at capturing the fine spectral features of samples and can be paired with various models for multi-indicator detection, tailored to specific application scenarios. However, several challenges remain. Firstly, HSI is susceptible to environmental factors such as lighting, shadows, and climatic conditions, which can alter the spectral signatures of samples. Secondly, the high spectral dimensionality and data redundancy complicate calculations, leading to the confusion of normal spectra, background content, and abnormal spectra, potentially distorting the content. Additionally, the significant storage space required for image data poses logistical challenges.

To address these issues, future developments in the HSI non-destructive testing technology should focus on upgrading the hardware, refining the detection model’s accuracy and robustness, reducing the spectral dimensions, and simplifying the data and calculations. These advancements will further enhance the practicality and effectiveness of HSI technology in diverse applications.

#### 2.1.3. Raman Spectroscopy Technology

Raman spectroscopy is a sophisticated light scattering technique that provides a wealth of information about a sample’s chemical composition and structure. This technique operates by exposing a sample to high-intensity incident light from a laser source, which results in two types of scattering: elastic and inelastic. Both of these scattering phenomena are collectively known as Raman scattering. The resulting Raman spectrum, which is based on the interaction between light and the chemical bonds within the material, is typically composed of multiple Raman peaks. Each peak corresponds to a specific wavelength position and intensity of the Raman scattered light [40,41]. These peaks offer researchers valuable insights into the sample, such as chemical structure identification, pH value determination, assessment of sample crystallinity, shear force measurement, and the detection of impurities, making Raman spectroscopy a versatile analytical tool [42,43,44]. The non-destructive nature of Raman spectroscopy, characterized by its straightforward operation, rapid measurement time, high sensitivity, and the ability to analyze samples without causing damage, ensures that its development prospects remain stable and ever improving.

Capitalizing on the strengths of Raman spectroscopy, numerous scholars worldwide have explored its application in the non-destructive testing for meat tenderness. As far back as 2004, researchers utilized Raman spectroscopy to investigate how proteins reflect the tenderness of beef. They designed and implemented a Raman spectroscopy-based non-destructive detection system, discovering that the ratio of α-helix to β-sheet structures in proteins and the hydrophobicity of muscle fibers are critical factors influencing the tenderness of beef. By employing the Partial Least Squares (PLS) algorithm for spectral data analysis, a strong correlation between Raman spectral data and beef tenderness was established, with an R^2^ value of 0.65 [45]. This finding underscores the significant potential of Raman spectroscopy in predicting meat tenderness.

Bauer et al. employed a 671 nm Raman system to non-destructively test the tenderness of beef. They scanned spectral data in the 340~2100 cm^−1^ range and preprocessed the data using Extended Multiplication Scattering Correction (EMSC). They then constructed a prediction model based on PLSR and PLS-DA algorithms. The PLS-DA model demonstrated the highest accuracy, reaching 80%, with classification accuracy improving as the set threshold increased [46]. This outcome confirms the feasibility of developing Raman spectroscopy-based non-destructive testing equipment for meat tenderness and offers valuable insights for researchers in equipment development.

Building upon the handheld Raman microprobe, a lamb tenderness data collection system was designed, as illustrated in Figure 4. Savitzky–Golay (SG) smoothing was used for spectral data preprocessing, while PLS was applied for modeling and analysis. The model fitting revealed a strong correlation between the Raman spectra generated by the detection and the actual sample tenderness, with R^2^ values of 0.79 and 0.86 for the two datasets, respectively, indicating a robust predictive capability [47]. The development of more convenient equipment signifies technological advancement, and this finding confirms that utilizing portable devices can enhance the work efficiency, increase the number of testing samples, and thereby strengthen the data training for prediction models, improving their generalizability.

Cama-Moncunell et al. utilized a 780 nm wavelength laser with 120 mW power, a charge-coupled device (CCD) detector, and a universal platform sampling (UPS) with a 50 um gap aperture as their Raman spectroscopy data acquisition platform for beef tenderness detection. They recorded Raman intensities within the 250~3381 cm^−1^ shift range and preprocessed the data using five different methods, including SG. They iteratively selected the most crucial Raman bands using PLS combined with cross-validation (CV) and projection variable importance (VIP) to fit a new PLS model. Unfortunately, the model’s prediction results were suboptimal, with the highest correlation coefficient reaching only 0.48. The low accuracy may stem from changes in the protein’s secondary and tertiary structures, which can obscure the α-helix signal, leading to unclear spectra and mixed bands. Additionally, the tryptophan signal in the 760~880 cm^−1^ band of the tertiary structure does not contribute to the prediction of tenderness, complicating the detection process. The presence of unrelated signals in the model may also contribute to its low accuracy [48]. This study’s contribution lies in its examination of Raman spectral bands for detecting WBSF and in summarizing the reasons behind the low accuracy of their detection model, providing valuable lessons for future researchers to avoid similar pitfalls.

Despite its extensive applications in non-destructive testing, Raman spectroscopy has certain limitations. Firstly, many compounds lack Raman activity, and those that possess it may emit fluorescence at specific excitation frequencies, which can overshadow Raman signals and impede analysis. Secondly, nonlinear curves are common in Fourier transform spectroscopy analysis. Thirdly, optical system parameters can influence overlapping vibrational peaks and variations in Raman scattering intensities. Addressing these challenges requires the enhancement of the Raman signal strength, bolstering the robustness of detection models, and improving the stability and fault tolerance of Raman spectroscopy systems, which are now key research focal points.

### 2.2. Airflow–Optical Fusion Detection Technology

Airflow–optical fusion detection technology represents a novel frontier in non-destructive testing, offering a unique approach to assess the quality characteristics of meat products. The fundamental premise of this technology is to utilize gas as a medium, employing pressure devices like pumps to compress the gas. By altering the cross-sectional area of a flow pipeline, the gas is transformed into a high-velocity jet that emanates from a small-diameter straight-hole nozzle in a near-radial pattern. When a test sample is positioned beneath the nozzle, the force exerted by the high-speed airflow can induce surface deformation. Concurrently, optical sensors are deployed to monitor the sample’s deformation, yielding a series of test data that can be analyzed for non-destructive evaluation of meat attributes such as its tenderness and freshness [49,50,51,52]. This technology has garnered extensive research and application, with airflow laser technology and airflow-structured light three-dimensional (3D) vision technology being particularly noteworthy.

Lee et al. devised an innovative laser blowing system that harnesses airflow–optical fusion detection technology to gauge the tenderness of chicken meat. By applying a high-speed airflow to the chicken meat, a laser sensor positioned above the airflow nozzle simultaneously recorded detection data. This data were then modeled using Ordinary Least Squares (OLS) and Partial Least Squares (PLS) algorithms, yielding promising results with a resolution accuracy of up to 85%. Interestingly, this system was found to be particularly adept at detecting tender meat, likely due to the low airflow pressure that is insufficient to cause deformation in tougher meat [53]. In contrast, spectral technology may encounter overlapping spectral signals when distinguishing between tender and tough meat, necessitating complex data preprocessing to mitigate this issue. This observation suggests that, in the realm of tenderness detection, airflow–optical fusion technology possesses greater developmental potential than single optical technology.

Lou et al. explored the use of airflow in conjunction with structured-light 3D vision technology for the rapid, non-destructive detection of beef tenderness. A structured-light 3D camera was used to capture the three-dimensional point cloud deformation data on the surface of beef following the application of an airflow. Employing algorithms such as point cloud segmentation, down sampling, and rotation, they extracted nine features from the beef surface, including six deformation and three point cloud features. Their findings indicated that a model based on the Extreme Learning Machine (ELM) algorithm exhibited a strong predictive capability for beef tenderness, with a R of 0.8356. Moreover, the ELM model achieved an impressive correct classification accuracy of 92.96% for tender beef [54].

Xu et al. developed a non-destructive testing apparatus leveraging controllable airflow–laser detection technology to assess the tenderness of chicken meat, as depicted in Figure 5. They employed a Support Vector Machine (SVM) and global variable PLS algorithms, combined with Savitzky–Golay (SG) preprocessing, to construct models for dynamic and static detection modes, such as transient, creep recovery, and stress relaxation. Their results indicated that the dynamic mode outperformed the static mode, with the transient mode demonstrating superior performance in both qualitative and quantitative assessments. The correlation coefficients for the validation set were an impressive 0.95 and 0.913, respectively [55].

Lu et al. introduced a rapid, non-destructive testing method for beef tenderness, integrating airflow pulses with structured-light 3D imaging to address the limitations of traditional tenderness testing methods in terms of speed and accuracy. The detection system, shown in Figure 6, utilized a high-speed airflow to impact the beef surface and structured-light 3D imaging to capture the three-dimensional point cloud information from the concave area formed on the beef’s surface. Through denoising, point cloud segmentation, and surface fitting algorithms, they extracted various effective data from the concave area. Additionally, they established beef tenderness prediction models based on a Least Squares Support Vector Machine (LS-SVR), Backpropagation Neural Network (BPNN), and Generalized Regression Neural Network (GRNN). The GRNN model emerged as the most accurate predictor, with a correlation coefficient of 0.975 for the prediction set. Furthermore, the researchers utilized a GRNN neural network based on K-fold cross-validation to predict the tenderness levels of beef, achieving a perfect grading accuracy of 100% for tender beef [56].

Despite the promising potential of airflow–optical fusion detection technology in meat product quality assessment, several challenges remain. The current research predominantly focuses on establishing the correlations between meat product quality characteristics and prediction models, with less emphasis on the development of dedicated detection equipment. Moreover, existing equipment tends to be bulky and inconvenient to transport. The control methods for these systems are often limited, relying on hardware to regulate the airflow output. Once the airflow exits the nozzle, it becomes uncontrollable, leading to an entrainment effect with the surrounding air, resulting in an uneven force distribution on the sample surface and complicating the deformation area’s output signal. To address these issues, it is essential to expedite the research on equipment portability and miniaturization and to enhance the key structures and associated software of the detection systems. Improving the robustness and fault tolerance of the detection models will significantly boost the capabilities of airflow–optical fusion detection technology.

### 2.3. Nuclear Magnetic Resonance (NMR) Technology

NMR is an analytical tool that is both information-rich and non-destructive in nature. The fundamental principle of NMR is based on the behavior of atomic nuclei in a constant magnetic field, where they undergo a precession effect around the external magnetic field. By introducing a fixed frequency electromagnetic wave and adjusting the strength of the external magnetic field to match the precession frequency with the electromagnetic wave frequency, nuclear precession synchronizes with the electromagnetic waves, a phenomenon known as NMR. This technology exploits the intrinsic magnetic properties of specific atomic nuclei to absorb and emit energy in the form of electromagnetic waves. This process induces nuclear transitions and generates NMR signals that reveal the molecular structure, dynamic processes, and chemical reactions within the sample, enabling non-invasive, non-destructive, and quantitative analysis and research [57,58,59]. As NMR is a magnetic field imaging method devoid of radioactivity, it is inherently safe and poses no harm to researchers. In recent years, NMR technology, characterized by its low cost, rapid non-destructive testing capabilities, and precision, has seen widespread application in the food, agriculture, and industrial sectors.

In their study, Kelly et al. employed NMR technology to investigate the migration and distribution characteristics of myofibrillar water in muscle tissues. They used a desktop low-field Nuclear Magnetic Resonance (LF-NMR) transverse relaxation (T2) instrument to characterize the distribution and fluidity of water in meat, as well as the structural characteristics that directly influence water retention capacity (WHC). Their research demonstrated that the T2 relaxation time can indicate meat’s water-holding capacity. Given that water content is a critical factor in meat tenderness studies, this work has established the feasibility of using NMR technology for the non-destructive testing of meat tenderness [60].

Focusing on Tan lamb as their subject, Ma et al. applied LF-NMR technology to study the water distribution and migration in cold and fresh Tan lamb during its storage to analyze and differentiate the tenderness levels. Using an NMR imaging analyzer, they sampled and imaged the lamb. Post-image analysis with a curve regression model revealed a highly significant correlation between the transverse relaxation time T2 and shear force, with a correlation coefficient of −0.996 (*p* < 0.01) and a fitting regression coefficient of 0.942. Similarly, the total peak area A exhibited a highly significant negative correlation with shear force, with a correlation coefficient of −0.991 (*p* < 0.01) and a fitting regression coefficient of 0.960 [61].

Fabíola et al. explored the use of time-domain Nuclear Magnetic Resonance (TD-NMR) technology for beef tenderness detection. They collected attenuation signals from Carr-Purcell-Meiboom-Gill (CPMG) and Continuous Wave-Free Precession (CWFP) sequences using a TD-NMR spectrometer and constructed a multiple regression model using Partial Least Squares (PLS) to fit the data. Their results indicated a strong correlation between CPMG tenderness data and the reference data, with r > 0.65. The study also highlighted that each beef sample could be measured in less than 1 s, confirming that this detection method not only offers good predictive performance but also boasts an extremely high detection efficiency [62]. TD-NMR detection technology provides researchers with a non-destructive and reliable analysis method, which can inform the design and manufacture of more user-friendly NMR non-destructive testing equipment.

While NMR technology is a convenient and swift detection method widely utilized in the medical field, its application in agriculture and food industries is less prevalent. There are several reasons for this discrepancy: 1. The application research in the food industry lacks a systematic approach, and there is a scarcity of detection data. 2. The NMR testing of different agricultural products necessitates specific research for each product, which is time-consuming. 3. The cost of the NMR-related equipment is relatively high, which to some extent curtails the advancement of NMR technology. To overcome these challenges, it is essential to intensify the data collection efforts, establish a more comprehensive NMR spectral database, and conduct research into low-cost, portable non-destructive testing equipment. These initiatives will be pivotal in propelling the development and broader adoption of NMR technology in various fields.

## 3. Research Development Trends and Prospects

As technology continues to evolve, the rapid non-destructive testing technologies for meat tenderness mentioned earlier are poised to become pivotal in the field. Table 1 encapsulates the diverse applications of each technology, showcasing their potential for enhancing the quality assessment of meat products. However, several challenges remain in developing and implementing these rapid, non-destructive testing technologies.

Firstly, technologies such as near-infrared spectroscopy, hyperspectral imaging, and nuclear magnetic resonance face limitations due to the bulkiness of the equipment, high costs, and portability issues, which hinder their widespread adoption and application. Secondly, the existing databases and models are often tailored to specific meat types, lacking the versatility to generalize the quality characteristics across various meat categories. This necessitates a targeted approach to data analysis, which limits the universality of the models.

To address these challenges, a multifaceted strategy is required. Firstly, there is a need to foster the integration and exploration of technologies, enhancing the capacity to merge multiple technologies into more precise and stable testing equipment. Secondly, upgrading and transforming the key components of the current testing equipment will improve the accuracy of meat tenderness assessments. Thirdly, intensifying research into the intelligence, miniaturization, and efficient data processing capabilities of the testing equipment is crucial for increasing the testing efficiency.

Additionally, establishing a comprehensive, multi-source, and open non-destructive testing database for meat tenderness is essential. This database would facilitate more robust data comparison and enhance the overall testing capabilities. Fifthly, a holistic summary of existing testing models, coupled with the integration of multi-source information, will pave the way for the development of a more universal and robust non-destructive testing model for meat tenderness.

The rapid advancement of big data and artificial intelligence (AI) has led to significant improvements in computing power and algorithms. Currently, the detection methods are often confined to specific algorithms, and the models developed are relatively singular. However, by harnessing AI, these limitations can be transcended. AI can analyze and compare original detection data with existing algorithms and models, distilling from vast datasets the most suitable algorithms and models for non-destructive testing. By recording these findings in a database and iteratively refining the AI, there is an expectation of establishing detection models with greater universality and stability.

Over time, as the data accumulate, AI may even devise previously unknown algorithms and models that are better suited for non-destructive testing. This development is instrumental in simplifying the intelligence required for non-destructive testing and holds immense potential for the future.

With the accelerating pace of technological innovation and the growing focus on intelligent instrumentation research and development globally, it is anticipated that in the near future, the convergence of rapid non-destructive testing and AI technology will lead to significant enhancements in the related instruments and methodologies. This integration will not only improve the efficiency and accuracy of meat tenderness assessments but also expand the applicability of these technologies across various industries, marking a new era in quality control and assurance.
foods-13-01512-t001_Table 1Table 1Research on tenderness based on non-destructive testing technology.TechnologyObject and  IndicatorsData ProcessingValidation Methods and ParametersNIRmutton tendernessMSC + PLSThe prediction accuracy of the model based on PLS reaches 0.96329 [24].pork tendernessMSC, 1st Der + PLSThe PLS model prediction accuracy is 78% [26].meat tendernessSNV, MSC, SG + PLSThe prediction accuracy of the model is average, and the RPD value is only occasionally greater than 2 [27]. beef tendernessMLRThe MLR model predicts a correlation coefficient of 0.806 [28].HSIbeef tendernessPCAThe validation set prediction accuracy of the linear discriminant model is 75% [33].beef tendernessPLS-DAThe recognition accuracy of the local model based on PLS-DA is 72% [34].beef tendernessGA + PCA-LDAThe recognition accuracy of the PLS-LDA model is 94.44% [35].mutton tendernessSG + PLSRThe correlation coefficient of the PLSR model for predicting the prediction set is 0.89 [36].chicken tendernessSG + PLSRThe constructed PLSR model has the best predictive performance, with a correlation coefficient of 0.94 in the prediction set [37].mutton tendernessMSC + BPNNThe BPNN model has the best predictive performance, with a correlation coefficient of 0.85 for the prediction set [38].mutton tendernessMSC, de rendering, baseline, SNV, normalize, SG + OS-IvISSA-PLSSRThe correlation coefficient of the OS IvISSA-PLSSR tenderness prediction model prediction set is 0.79 [39].Ramanbeef tendernessPLSThe R^2^ of the PLS beef tenderness prediction model is 0.65 [45]beef tendernessEMSC + PLS-DAThe accuracy of the PLS-DA prediction model reaches 80% [46].mutton tendernessSG + PLSThe predictive correlation coefficients (R^2^) of the PLS model for two sets of data are 0.79 and 0.86, respectively [47].Airflow-optical fusion detectionchicken tendernessOLS, PLSThe accuracy of the detection model is 85% [53].beef tendernessELMThe detection model has good predictive performance, with a correlation coefficient of 0.8356, and a discrimination rate of 92.96% for tender beef [54].Airflow-optical fusion detectionchicken tendernessSG + global variable PLSThe correlation coefficients of the constructed transient modal model in qualitative and quantitative validation sets are 0.95 and 0.913, respectively [55]beef tendernessGRNN, K-foldThe GRNN neural network based on K-fold cross-validation has a good grading effect on tender beef, with a discrimination effect of 100% [56].NMRmutton tendernesscurve regressionT2 is negatively linked to shear force (R = −0.996, *p* < 0.01), and A is positively linked (R = 0.960) [61].beef tendernessCPMG + PLSThe CPMG tenderness data detection effect is good, with r > 0.65 [62].

## Figures and Tables

**Figure 1 foods-13-01512-f001:**
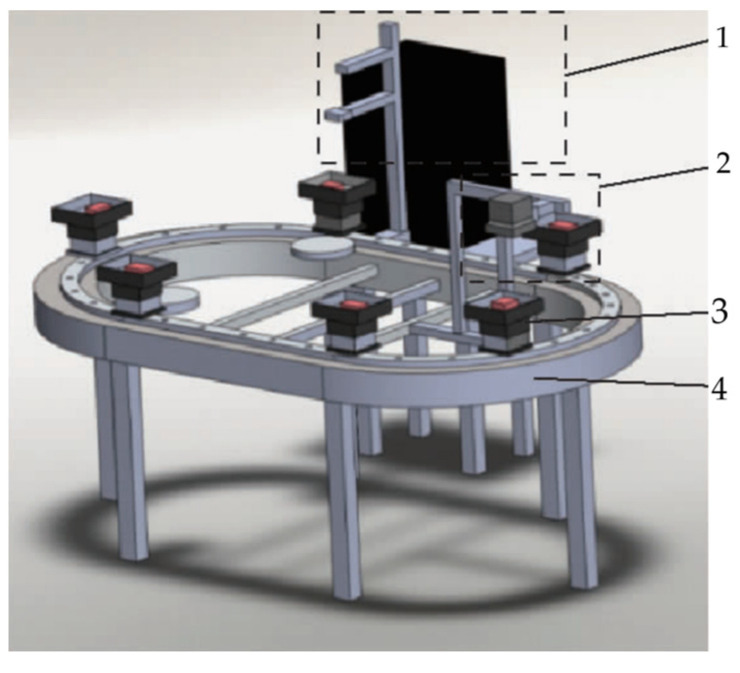
Working principle diagrams of detection device. This figure shows the main structure of the device. Note: 1. Spectral acquisition unit; 2. Distance measuring unit; 3. Sample placement table; 4. Sample transmission unit.

**Figure 2 foods-13-01512-f002:**
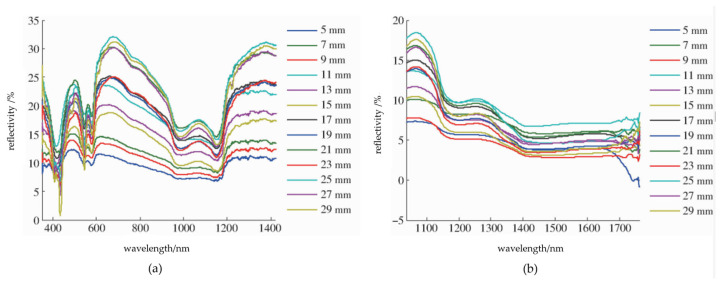
Original spectrum at different distances of the first band and second band. This figure shows the analysis process of the raw data. Note: (**a**) First band; (**b**) Second band.

**Figure 3 foods-13-01512-f003:**
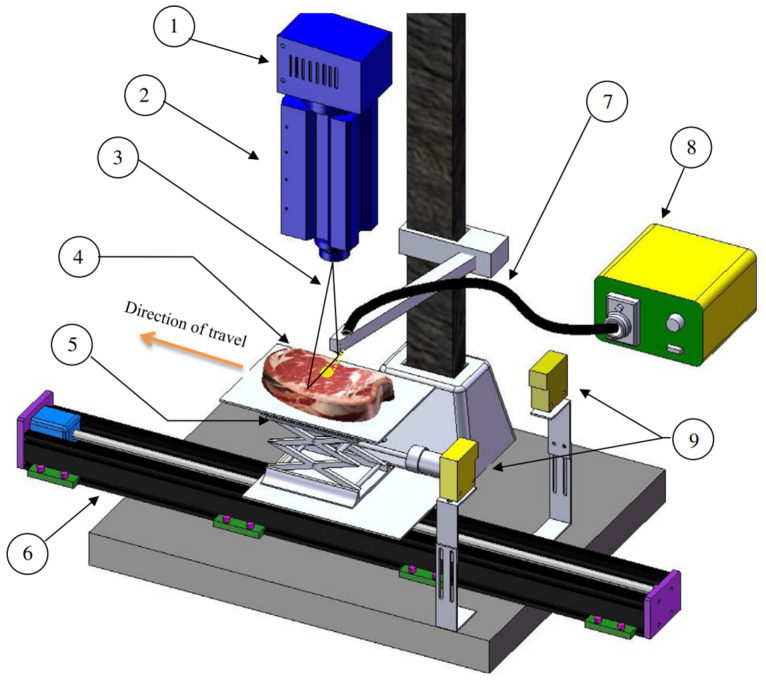
Schematic diagram of hyperspectral imaging system. This system is used to collect hyperspectral scattering images. Note: 1. InGaAs camera; 2. spectrograph; 3. FOV of line scan camera offset 5 mm from center of incident beam of light; 4. steak sample; 5. automated vertical stage; 6. linear slide, which moved the sample under the line scan in the direction of travel; 7. incident fiber optic cable; 8. light source tungsten halogen lamp; 9. photoelectric switch.

**Figure 4 foods-13-01512-f004:**
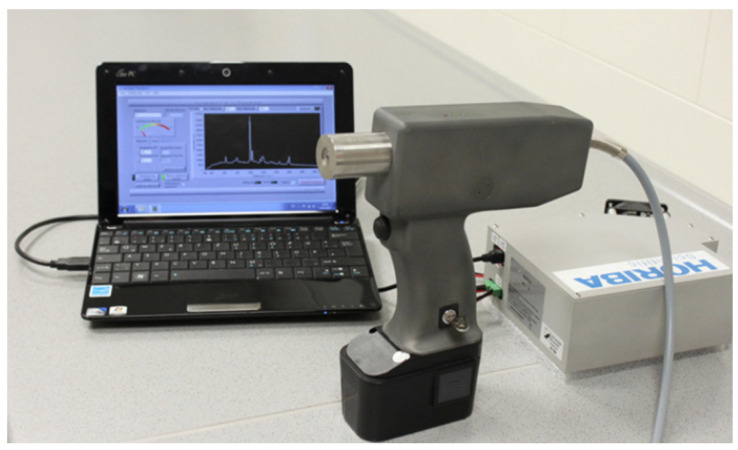
Portable Raman system. This figure shows the composition of the portable Raman system. Note: Handheld Raman device (**center**), miniaturized spectrometer (**right**), and netbook (**left**).

**Figure 5 foods-13-01512-f005:**
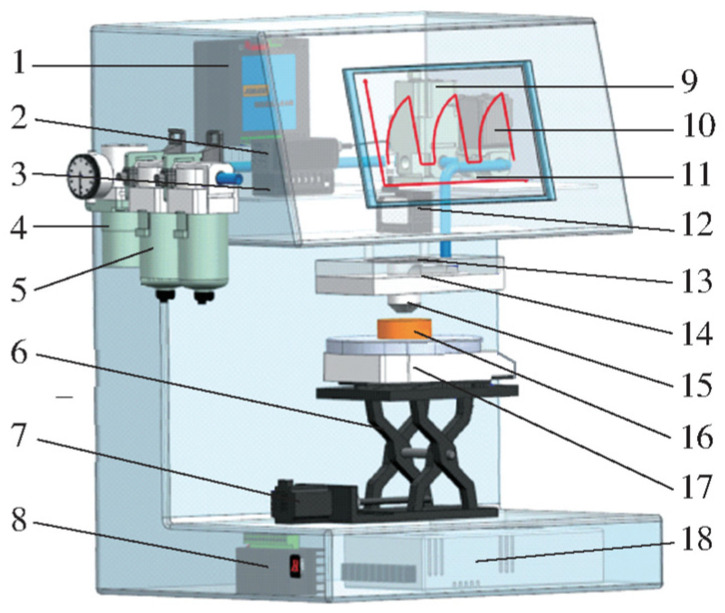
Structure diagram of controlled laser–airflow detection system. This figure shows us in detail all the components of the detection system. Note: 1. Analog input output conversion module; 2. Laser signal amplifier; 3. Laser displacement sensor A/D conversion module; 4. Pressure regulating valve; 5. Two-stage air filtration; 6. Lifting table; 7. Stepper motor; 8. Stepper motor driver; 9. Electrical proportional valve; 10. Electromagnetic valve; 11. ARM embedded integrated machine; 12. Laser displacement sensor; 13. Quartz window; 14. Air chamber; 15. Nozzles; 16. Samples; 17. Electromagnetic force balance sensor; 18. Power switch.

**Figure 6 foods-13-01512-f006:**
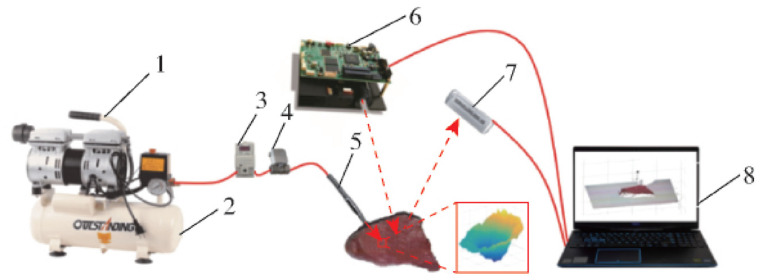
Air-puff and structured-light system. This figure shows the overall structure of the detection system. Note: 1. Small air compressor; 2. Air storage tank; 3. SMC proportional valve; 4. Pneumatic solenoid valve; 5. Ventilation pipeline and nozzle; 6. DLP digital projector; 7. Camera; 8. Portable computer.

## Data Availability

No new data were created or analyzed in this study. Data sharing is not applicable to this article.

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
