# Peer review of "Rapid Non-Destructive Detection Technology in the Field of Meat Tenderness: A Review"

_foods, 2024, doi:10.3390/foods13101512_

Round 1

Reviewer 1 Report

Comments and Suggestions for Authors

Rapid Non-Destructive Detection Technology in the Field of Meat Tenderness:A Review

In the meat industry, several advanced non-destructive techniques have emerged as precise, efficient, fast, and simple or non-existent ways to prepare samples. This article reviews five techniques for evaluating meat tenderness, one of the most important factors associated with meat quality.

This article is well-written and shows the authors' strong command of the subject. The clarity of the arguments and logical flow of ideas make it engaging and easy to follow. The authors demonstrate a deep understanding of the topic and effectively support their claims with relevant evidence and examples from the scientific literature. The language is precise but accessible, serving both specialists and general readers. Overall, this article represents a relevant contribution to food and meat science.

Some detailed comments are below:

Line 13 Meat serves as a vital “change with” Meat is a vital 

17 The tenderness of meat is “change with” The tenderness of the meat is

18 testing, a method that is inherently destructive, inefficient, and results in considerable sample wastage “change with” testing, which is inherently destructive and inefficient and results in considerable sample wastage

22 This paper delves into the application of five advanced “change with” This paper delves into applying five advanced

32 recommendations aimed at guiding “change with” the recommendations to guide the

45 tenderness of meat is “change with” tenderness of the meat is

68 with their use, and highlight “change with” with their use, and importantly, highlight

92 parameters such “change with” parameters, such

128 In tenderness detection of meat products, NIR spectroscopy coupled with successive projection algorithm (SPA) has yielded promising results. “change with” NIR spectroscopy coupled with successive projection algorithm (SPA) has yielded promising results in tenderness detection of meat products.

137 Integrating NIR reflectance spectroscopy with visible light, Balage et al. attempted pork tenderness detection. “change with” Integrating NIR reflectance spectroscopy with visible light, Balage et al. attempted pork tenderness detection.

164 without the need for any “change with” without needing any

175 The configured HSI system, as depicted in Figure 3, gathers data from samples “change with” Configured as depicted in Figure 3, the HSI system gathers data from samples

182 Balage et al. “change with” Balage et al. [26]. Check the manuscript for this issue.

200 and for evaluating beef tenderness comprehensively. “change with” and comprehensively evaluating beef tenderness.

231 models are crucial “change with” models is crucial

329 Thirdly, overlapping vibrational peaks and variations in Raman scattering intensities can be influenced by optical system parameters. “change with” Thirdly, optical system parameters can influence overlapping vibrational peaks and variations in Raman scattering intensities.

441 Their research demonstrated that the T2 relaxation time can serve as an indicator of meat's water holding capacity. “change with” Their research demonstrated that the T2 relaxation time can indicate meat's water holding capacity.

480 However, several challenges remain in the development and implementation of these of these rapid “change with” However, several challenges remain in developing and implementing these rapid, 

Comments on the Quality of English Language

Only minor editing of the English language is required.

Author Response

Dear Reviewer:

Thank you very much for giving us an opportunity to make a revision for our manuscript entitled "Rapid Non-Destructive Detection Technology in the Field of Meat Tenderness:A Review". We have studied comments and revised everything carefully as you suggested. These revisions are highlighted by using coloured (red) text.

Our responses to you are as follows:

Line 13 Meat serves as a vital “change with” Meat is a vital 

Thank you for your suggestion. Having taken into account the feedback from other reviewers, adjustments have been implemented in this section.

17 The tenderness of meat is “change with” The tenderness of the meat is

Thank you for your suggestion. Having taken into account the feedback from other reviewers, adjustments have been implemented in this section.

18 testing, a method that is inherently destructive, inefficient, and results in considerable sample wastage “change with” testing, which is inherently destructive and inefficient and results in considerable sample wastage

Thank you for your suggestion. The relevant contents have been added to the revised manuscript (Lines 13-14 in Abstract). The changes have been highlighted with revised format.

22 This paper delves into the application of five advanced “change with” This paper delves into applying five advanced

Thank you for your suggestion. The relevant contents have been added to the revised manuscript (Lines 18 in Abstract). The changes have been highlighted with revised format.

32 recommendations aimed at guiding “change with” the recommendations to guide the

Thank you for your suggestion. The relevant contents have been added to the revised manuscript (Lines 28 in Abstract). The changes have been highlighted with revised format.

45 tenderness of meat is “change with” tenderness of the meat is

Thank you for your suggestion. Having taken into account the feedback from other reviewers, adjustments have been implemented in this section.

68 with their use, and highlight “change with” with their use, and importantly, highlight

Thank you for your suggestion. The relevant contents have been added to the revised manuscript (Lines 60 in chapter 1). The changes have been highlighted with revised format.

92 parameters such “change with” parameters, such

Thank you for your suggestion. The relevant contents have been added to the revised manuscript (Lines 80 in chapter 2). The changes have been highlighted with revised format.

128 In tenderness detection of meat products, NIR spectroscopy coupled with successive projection algorithm (SPA) has yielded promising results. “change with” NIR spectroscopy coupled with successive projection algorithm (SPA) has yielded promising results in tenderness detection of meat products.

Thank you for your suggestion. The relevant contents have been added to the revised manuscript (Lines 109-110 in chapter 2.1.1). The changes have been highlighted with revised format.

137 Integrating NIR reflectance spectroscopy with visible light, Balage et al. attempted pork tenderness detection. “change with” Integrating NIR reflectance spectroscopy with visible light, Balage et al. attempted pork tenderness detection.

Thank you for your suggestion. The relevant contents have been added to the revised manuscript (Lines 118-119 in chapter 2.1.1). The changes have been highlighted with revised format.

164 without the need for any “change with” without needing any

Thank you for your suggestion. The relevant contents have been added to the revised manuscript (Lines 144-145 in chapter 2.1.2). The changes have been highlighted with revised format.

175 The configured HSI system, as depicted in Figure 3, gathers data from samples “change with” Configured as depicted in Figure 3, the HSI system gathers data from samples

Thank you for your suggestion. The relevant contents have been added to the revised manuscript (Lines 158 in chapter 2.1.2). The changes have been highlighted with revised format.

182 Balage et al. “change with” Balage et al. [26]. Check the manuscript for this issue.

Thank you for your suggestion. Upon thorough review of the text, it has been verified that the authors of this particular section are identical to those cited in reference [26]. To differentiate between the two works, annotations precede the authors' names—refer to lines 118 in chapter 2.1.1 and lines 165 in chapter 2.1.2.

200 and for evaluating beef tenderness comprehensively. “change with” and comprehensively evaluating beef tenderness.

Thank you for your suggestion. The relevant contents have been added to the revised manuscript (Lines 183 in chapter 2.1.2). The changes have been highlighted with revised format.

231 models are crucial “change with” models is crucial

Thank you for your suggestion. The relevant contents have been added to the revised manuscript (Lines 214 in chapter 2.1.2). The changes have been highlighted with revised format.

329 Thirdly, overlapping vibrational peaks and variations in Raman scattering intensities can be influenced by optical system parameters. “change with” Thirdly, optical system parameters can influence overlapping vibrational peaks and variations in Raman scattering intensities.

Thank you for your suggestion. The relevant contents have been added to the revised manuscript (Lines 311-312 in chapter 2.1.3). The changes have been highlighted with revised format.

441 Their research demonstrated that the T2 relaxation time can serve as an indicator of meat's water holding capacity. “change with” Their research demonstrated that the T2 relaxation time can indicate meat's water holding capacity.

Thank you for your suggestion. The relevant contents have been added to the revised manuscript (Lines 422-423 in chapter 2.3). The changes have been highlighted with revised format.

480 However, several challenges remain in the development and implementation of these of these rapid “change with” However, several challenges remain in developing and implementing these rapid, 

Thank you for your suggestion. The relevant contents have been added to the revised manuscript (Lines 461-462 in chapter 3). The changes have been highlighted with revised format.

At last, sincerely thank you for the comments.

Sincerely,

Yanlei Li

E-mail: liyanlei2021@163.com

Reviewer 2 Report

Comments and Suggestions for Authors

The article titled „Rapid Non-Destructive Detection Technology in the Field of Meat Tenderness: A Review” has enormous scientific potential.

Abstract:

Page 1, Line 18: Traditionally, tenderness has been assessed through shear force testing, a method that is inherently destructive, inefficient, and results in considerable sample wastage.

In my opinion, this method cannot be called ineffective. There are numerous scientific studies confirming the effectiveness of shear force in determining meat tenderness. This method is effective and practiced all over the world.

Page 2, line 77: sophisticated instruments - incorrectly used term

Page 2, line 87: Robust- incorrectly used term

Page 13, table 1: T2 is negatively linked to shear force (R=-0.996) and A is positively linked (R=0.960) [61] - not marked in yellow

Author Response

Dear Reviewer:

Thank you very much for giving us an opportunity to make a revision for our manuscript entitled "Rapid Non-Destructive Detection Technology in the Field of Meat Tenderness:A Review". We have studied comments and revised everything carefully as you suggested. These revisions are highlighted by using coloured (orange) text.

Our responses to you are as follows:

Abstract:

Page 1, Line 18: Traditionally, tenderness has been assessed through shear force testing, a method that is inherently destructive, inefficient, and results in considerable sample wastage.

In my opinion, this method cannot be called ineffective. There are numerous scientific studies confirming the effectiveness of shear force in determining meat tenderness. This method is effective and practiced all over the world.

Thank you for your suggestion. The relevant contents have been added to the revised manuscript (Lines 14-16 in Abstract). The changes have been highlighted with revised format.

Page 2, line 77: sophisticated instruments - incorrectly used term

 Thank you for your suggestion. The relevant contents have been added to the revised manuscript (Lines 67 in in chapter 2). The changes have been highlighted with revised format.

Page 2, line 87: Robust- incorrectly used term

Thank you for your suggestion. Having taken into account the feedback from other reviewers, adjustments have been implemented in this section.

Page 13, table 1: T2 is negatively linked to shear force (R=-0.996) and A is positively linked (R=0.960) [61] - not marked in yellow

 Thank you for your suggestion. We have made changes to the highlighted parts in the table and checked them throughout the entire text.

At last, sincerely thank you for the comments.

Sincerely,

Yanlei Li

E-mail: liyanlei2021@163.com

Reviewer 3 Report

Comments and Suggestions for Authors

The manuscript provides interesting information in the field of meat tenderness investigate by use non-destructive technologies. However, in my opinion, some revisions to the text are necessary. I hope the manuscript after careful revisions meet high standards of the journal.

Abstract

Lines 13-17: I suggest remove.

Lines 33-34: „thereby contributing to the ongoing improvement of the meat industry in China and beyondI suggest remove”. I suggest remove.

Keywords

I suggest write as: meat; tenderness; non-destructive testing.

Main text:

Throughout the manuscript:

Please explain the abbreviations used in the subchapter titles.

Please insert relevant number in square brackets after the reference.

Lines 40-41: I suggest remove.

Lines 45-46: Please add the references and definitions of meat tenderness.

Lines 71-73: I suggest remove.

Lines 87-88: I suggest remove.

Lines 98-105: I suggest remove.

Line 174: „In previous studies ...”. Please insert the references.

Please add short conclusions

References

All Latin names should be italicized.

Author Response

Dear Reviewer:

Thank you very much for giving us an opportunity to make a revision for our manuscript entitled "Rapid Non-Destructive Detection Technology in the Field of Meat Tenderness:A Review". We have studied comments and revised everything carefully as you suggested. These revisions are highlighted by using coloured (blue) text.

Our responses to you are as follows:

Abstract

Lines 13-17: I suggest remove.

Thank you for your suggestion. The relevant contents have been revised.

Lines 33-34: thereby contributing to the ongoing improvement of the meat industry in China and beyondI suggest remove”. I suggest remove.

Thank you for your suggestion. The relevant contents have been revised.

Keywords

I suggest write as: meat; tenderness; non-destructive testing.

Thank you for your suggestion. The relevant contents have been revised.

Main text:

Throughout the manuscript:

Please explain the abbreviations used in the subchapter titles.

Thank you for your suggestion. The abbreviations used in the chapter headings have been explained in the article.

Please insert relevant number in square brackets after the reference.

Thank you for your suggestion. The relevant references have been cross referenced within the article, but due to submission system reasons, cross referencing may lead to errors during transmission.

Lines 40-41: I suggest remove.

Thank you for your suggestion. The relevant contents have been revised.

Lines 45-46: Please add the references and definitions of meat tenderness.

Thank you for your suggestion. The relevant contents have been revised.

Lines 71-73: I suggest remove.

Thank you for your suggestion. The relevant contents have been revised.

Lines 87-88: I suggest remove.

Thank you for your suggestion. The relevant contents have been revised.

Lines 98-105: I suggest remove.

Thank you for your suggestion. The relevant contents have been revised.

Line 174: “In previous studies ...”. Please insert the references.

Please add short conclusions

Thank you for your suggestion. The relevant contents have been added to the revised manuscript (Lines 155-157 in chapter 2.1.2). The changes have been highlighted with revised format.

References

All Latin names should be italicized.

Thank you for your suggestion. The relevant contents have been revised.

At last, sincerely thank you for the comments.

Sincerely,

Yanlei Li

E-mail: liyanlei2021@163.com